# Surface Persistence of Trace Level Deposits of Highly Energetic Materials

**DOI:** 10.3390/molecules24193494

**Published:** 2019-09-26

**Authors:** Leonardo C. Pacheco-Londoño, José L. Ruiz-Caballero, Michael L. Ramírez-Cedeño, Ricardo Infante-Castillo, Nataly J. Gálan-Freyle, Samuel P. Hernández-Rivera

**Affiliations:** 1R3-C Research and Education Component of ALERT DHS Center of Excellence for Explosives Research, Department of Chemistry, University of Puerto Rico, Mayaguez Campus, Mayaguez, PR 00681, USA; jose.ruiz9@upr.edu (J.L.R.-C.); michael.l.ramirez@gmail.com (M.L.R.-C.); nataly.galan@unisimonbolivar.edu.co (N.J.G.-F.); 2School of Basic and Biomedical Sciences, Universidad Simón Bolívar, Barranquilla, 080020 Atlantico, Colombia; 3Joseph Smith & Sons Inc., Capitol Heights, MD 20743, USA; 4Department of Chemistry and Biochemistry, George Mason University, Fairfax, VA 22030, USA; 5Department of Physics-Chemistry, University of Puerto Rico, Arecibo, PR 00614, USA; ricinfante@gmail.com

**Keywords:** sublimation, explosive, FTIR, thermogravimetric analysis, grazing angle

## Abstract

In the fields of Security and Defense, explosive traces must be analyzed at the sites of the terrorist events. The persistence on surfaces of these traces depends on the sublimation processes and the interactions with the surfaces. This study presents evidence that the sublimation process of these traces on stainless steel (SS) surfaces is very different than in bulk quantities. The enthalpies of sublimation of traces of four highly energetic materials: triacetone triperoxide (TATP), 2,4-dinitrotoluene (DNT), 2,4,6-trinitrotoluene (TNT), and 1,3,5- trinitrohexahydro-s-triazine (RDX) deposited on SS substrates were determined by optical fiber coupled-grazing angle probe Fourier Transform Infrared (FTIR) Spectroscopy. These were compared with enthalpies of sublimation determined by thermal gravimetric analysis for bulk amounts and differences between them were found. The sublimation enthalpy of RDX was very different for traces than for bulk quantities, attributed to two main factors. First, the beta-RDX phase was present at trace levels, unlike the case of bulk amounts which consisted only of the alpha-RDX phase. Second, an interaction between the RDX and SS was found. This interaction energy was determined using grazing angle FTIR microscopy. In the case of DNT and TNT, bulk and traces enthalpies were statistically similar, but it is evidenced that at the level of traces a metastable phase was observed. Finally, for TATP the enthalpies were statistically identical, but a non-linear behavior and a change of heat capacity values different from zero was found for both trace and bulk phases.

## 1. Introduction

The residence time of a highly energetic material (HEM) on a surface can be defined as the time that the material persists on the surface after its deposition. The concept is essential for the development of samples and standards for trace detection systems [1,2,3,4,5,6,7,8,9,10,11,12]. Aside from adhesion considerations, the residence time mainly depends on the vapor pressure of the compound and surface–HEM interactions. The vapor pressure of a HEM and its interaction with any given surface can be characterized in terms of the desorption energy and the sublimation enthalpy. The desorption energy ($\Delta_{des}U$) can be defined as the change in energy when a substance adsorbed on a surface is desorbed. The desorption of an adsorbed molecule is an elemental surface kinetics process and is a measure of the strength of the interaction between the surface and the adsorbed species [13]. The enthalpy of sublimation ($\Delta_{sub}H$) is the energy change when a compound changes from the solid phase to the gas phase. These enthalpies are present for a solid deposited on a surface. If the solid-surface interaction is small, $\Delta_{des}U$ is insignificant, and the sublimation is the main phenomenon. There are two general ways of calculating $\Delta_{sub}H$, i.e., directly and indirectly. In the direct method, a calorimeter is used to measure the heat exchanged during the change of phase. In the indirect determination, the vapor pressure, or a proportional parameter thereof, is measured at different temperatures and the enthalpy can be calculated by use of the Clausius–Clapeyron equation [14,15,16]. Various approaches can be taken to characterize the vapor pressures of materials. These include direct measurements with a manometer [17], the use of mass spectrometry to monitor the gas phase concentration of the species, measurement of sample volatilization by vacuum diffusion [18] (using a Knudsen cell), and boiling point determination under different pressures by differential scanning calorimetry. Several studies have reported that thermogravimetric analysis (TGA) is a rapid and convenient method to obtain vapor pressure curves and the enthalpies of sublimation and vaporization of volatile materials, such as active pharmaceutical ingredients and HEMs with different vapor pressures [19,20,21,22]. Sublimation enthalpies can also be measured by spectroscopic methods such as fluorescence. For example, Stefanov et al. used fluorescence monitoring to estimate the sublimation enthalpies of tetraphenylporphyrin, porphine, and Nile red, a fluorescent intracellular dye [23]. In the present study, the sublimation enthalpies of triacetone triperoxide (TATP), 2,4-dinitrotoluene (DNT), 2,4,6-trinitrotoluene (TNT), and 1,3,5-trinitrohexahydro-s-triazine (RDX) deposits on stainless steel (SS) substrates were obtained from mid-infrared (MIR) grazing angle probe fiber-optic (GAP)-coupled FTIR spectroscopy measurements, which were performed under isothermal conditions at different temperatures. The results were then compared with those obtained for the bulk samples by TGA. Furthermore, $\Delta_{des}U$ for RDX on SS was obtained by thermal desorption spectroscopy (TDS) measurements using grazing-angle objective (GAO) FTIR microspectroscopy.

## 2. Materials and Methods

### 2.1. Reagents

The reagents used in this research were acetone (CH${}_{3}$COCH${}_{3}$, 98%, Aldrich-Sigma Chemical Co., Milwaukee, WI, USA), isopropanol (99%, Aldrich-Sigma, Saint Louis, MO, USA), hydrogen peroxide (H${}_{2}$O${}_{2}$, 50% in water, Aldrich-Sigma), hydrochloric acid (HCl, 12 M, Merck, VWR, Inc., West Chester, PA, USA), sulfuric acid (H${}_{2}$SO${}_{4}$, 18 M, Merck, VWR), and dichloromethane (CH${}_{2}$Cl${}_{2}$, Aldrich-Sigma). Standard solutions of RDX (1000 ppm in acetonitrile, GC/MS primary standards grade) were obtained from Restek Corp. (Bellefonte, PA, USA) and from Chem Service, Inc. (West Chester, PA, USA). Crystalline samples of 2,4-DNT and TNT were purchased Chem Service, Inc.

### 2.2. Synthesis of TATP

Caution: TATP is a primary explosive sensitive to impact, friction, electric discharge, and flame. The synthesis and handling of TATP are dangerous operations that require safety precautions. TATP could not be purchased from chemical suppliers at the level of purity and amount required for the study. Samples were prepared in small quantities as needed without storing due to the high thermal instability of this powerful and highly unstable explosive. For the synthesis, 1 mL of 0.01 M HCl was mixed with 3 mL of peroxide, 2 mL of acetone, and 4 mL water. Crystals formed after 5 h. The crystals were then filtered and washed, first with cold distilled water and next with a small amount of cold methanol. The solid was then recrystallized from methanol.

### 2.3. Instrumentation

A grazing angle probe (GAP; Remspec Corp., Charlton, MA, USA) interfaced to a Vector-22 FTIR interferometer (Bruker Optics, Billerica, MA, USA) equipped with an external mercury cadmium telluride (MCT) MIR detector was used for the spectroscopic monitoring of analytes deposited on the test surfaces. The GAP head uses carefully aligned mirrors to deliver a MIR beam to the sample surface at the grazing angle (approximately 80${}^{{}^{\circ}}$ from the surface normal) [24]. The spectrometer was coupled to the GAP by a MIR-transmitting fiber-optic cable [25,26]. The 1.5 m fiber was made from a chalcogenide (As–Se–Te) optical glass bundle that transmits throughout the MIR region except for a strong H-Se absorbance band at 2200 cm${}^{-1}$. The GAP beam was focused on a solid surface that was in thermal contact with an aluminum block equipped with a temperature controlled water bath. Samples were placed on SS plates on top of this surface. The experimental setup is illustrated in Figure 1.

The MIR beam was reflected in an elliptical shape from the metal surface. The size of the ellipse along the major axis was ≈16 cm and along the minor axis was ≈3 cm. To measure this axis, the intensity of infrared in different position on gold surface was measure using a thermal camera, the beam intensity pattern on the surface was well described by a Gaussian distribution [27]. The behavior of the relative intensity ($I_{r}$) was measured and can be fitted as Equation (Equation 1):
(1)$$I_{r}=exp\left[-\left(0.10\pm 0.01\right)x^{2}-\left(3.3\pm 0.2\right)y^{2}\right]$$

Rewriting Equation (Equation 1),
(2)$${\left(\frac{x}{3.2}\right)}^{2}+{\left(\frac{y}{0.6}\right)}^{2}=-ln\left(I_{r}\right)$$

This expression describes an ellipse with the major axis measuring 3.4 cm and the minor axis measuring 0.6 cm. These values for the axes represent an ellipse that contains 63.2% of the infrared beam [28]. For 99% reflection of the MIR light on the surface, the ellipse had dimensions of 6.8 cm on the major axis and 1.2 cm on the minor axis Infrared microscopy in the MIR region was also used to characterize the samples. A Bruker Optics model IFS 66v/S spectrometer coupled to a Hyperion II IR microscope equipped with GAO was used. A computer-controlled motorized stage, a cryocooled MCT detector and a potassium bromide (KBr) beam splitter allowed sampling of areas with dimensions of 100 × 100 μm${}^{2}$. TGA was done using a model Q-500 (TA Instruments, New Castle, DE, USA) for all bulk measurements. A constant ultrahigh purity nitrogen flow of 40 mL/min was used to run the samples. Standard platinum sample holders were used. Aluminum pans on top of the platinum holders were used to contain the samples to a specific area. The TGA was temperature calibrated using the nickel Curie point (356 ${}^{{}^{\circ}}$C) apparent weight loss according to manufacturer-optimized procedures.

### 2.4. Sublimation Study by Thermal Analysis

TGA is a rapid and convenient method for obtaining vapor pressure curves and the enthalpies of sublimation and vaporization of volatile materials. However, TGA cannot be used for measuring the sublimation of thin layers because the limit of mass determination of a TGA apparatus is higher than the mass of a thin layer. For example, for a surface of 1 cm${}^{2}$ that contains 0.1 μg/cm${}^{2}$ of material, the total mass is 100 ng (0.1 μg), which is too low to be measured by the microbalance in a conventional TGA instrument [29]. TGA methods for measuring vapor pressure and sublimation enthalpies are based on the principle that sublimation, as well as evaporation, is a zero-order process. Thus, the mass loss under isothermal conditions must be constant [21,30]. The mathematical expression that correlates the vapor pressure originates from Langmuir’s work of 1913 [31]:
(3)$$\left(\frac{1}{area}\right)\left(\frac{dm}{dt}\right)=p\alpha_{\nu }{\left(\frac{M}{2\pi R_{g}T}\right)}^{0.5}$$
where (1/area)(dm/dt) is the rate of mass loss per unit area, *p* is the vapor pressure, *M* is the molecular mass of the evaporating compound, $R_{g}$ is the gas constant, *T* is the absolute temperature (K), and $\alpha_{\nu }$ is the vaporization coefficient. In a vacuum, $\alpha_{\nu }$ is assumed to be 1, but in a flowing gas, such as that used in TGA experiments, $\alpha_{\nu }$ can have different values. Rearranging the Langmuir equation results in
(4)$$p=k_{t}\nu_{s}$$
where k${}_{t}$=(1/area)((2πR${}_{g}$)${}^{0.5}$/$\alpha_{\nu }$) and $\nu_{s}$=(dm/dt)(T/M)${}^{0.5}$. If a compound is thermally stable and its vapor pressures at different temperatures are known, it is possible to correlate the vapor pressure to the mass loss rates obtained by TGA from which $k_{t}$ is obtained [32]. This can be used to determine the vapor pressure of other substances. The study of the sublimation kinetics of HEMs by TGA involves determining the rate of mass loss at several isothermal points over the temperature range of interest. In the present study, the mass of the samples was monitored under isothermal conditions for a minimum of 30 min. The temperature range was different for each explosive at data intervals of 5 ${}^{{}^{\circ}}$C, 1 ${}^{{}^{\circ}}$C, and 0.5 ${}^{{}^{\circ}}$C. For TATP, the temperature range was 20 to 65 ${}^{{}^{\circ}}$C; for 2,4-DNT, the range was 25–70 ${}^{{}^{\circ}}$C; for TNT, the range was 30 to 90 ${}^{{}^{\circ}}$C; and for RDX, the range was 50 to 125 ${}^{{}^{\circ}}$C. To determine the vapor pressure, the rate of mass loss for benzoic acid was measured every 2 ${}^{{}^{\circ}}$C from 22 to 90 ${}^{{}^{\circ}}$C. The value of $\nu_{s}$ was calculated, and the value of k${}_{t}$ was obtained by fitting the vapor pressures in the literature [33,34].

### 2.5. Sublimation Study by GAP

The sublimation of thin layers and trace amounts of HEMs was studied by FTIR-GAP. Samples were prepared by depositing the HEMs from liquid solutions, generating homogeneous distributions on the surfaces of the SS substrates after drying at room temperature. The morphologies of the residues of the HEMs on the SS surfaces were terraces for TATP, droplets for 2,4-DNT and TNT, and layers for RDX, as determined using optical microscopy at 100× magnification (see Appendix A). The initial concentrations varied depending on differences in the solubilities and vapor pressures of the materials. Aliquots of 20 μL of HEMs standard solutions were deposited on one side of the SS substrates, and then evenly distributed using a sample smearing method [26]. Isopropanol was used as the solvent for RDX, TNT, and 2,4-DNT, and dichloromethane was used for TATP because of its high vapor pressure, which means that a more volatile solvent is required. For the determination of thermodynamic properties, MIR spectra were recorded as a function of time at different temperatures. For TATP, initial surface loadings (Cs${}_{{}^{\circ}}$) of 25, 50, and 80 μg/cm${}^{2}$ were used between 14 and 36 ${}^{{}^{\circ}}$C. The Cs${}_{{}^{\circ}}$ values for 2,4-DNT were 2.8, 5.7, and 11.4 μg/cm${}^{2}$ and for TNT were 3.8, 7.6, and 11.4 μg/cm${}^{2}$. The temperatures studied were 23–60 ${}^{{}^{\circ}}$C for 2,4-DNT and 22, 30, 40, 50, 55 and 70 ${}^{{}^{\circ}}$C for TNT. For RDX, only $Cs_{{}^{\circ}}$ values of 0.7 and 1.4 μg/cm${}^{2}$ were studied at 22, 44, 65, 75, and 80 ${}^{{}^{\circ}}$C. For each temperature, the measurements were carried out in triplicate. The sublimation rate of RDX at room temperature (22 ${}^{{}^{\circ}}$C) was monitored over 258 days to consider the signal decay.

### 2.6. Desorption Energy

Due to the low vapor pressure of RDX, its desorption energy was also studied by TDS. A temperature-programmed method (TPM) was used. A first-order desorption rate ($r_{des}$) was obtained for the RDX on SS substrates. For a first-order rate, the value of surface concentration or surface loading (Cs) is proportional to the $r_{des}$ and corresponds to the simplest case of single molecules desorbing directly and independently from sites on the surface. The $r_{des}$ is related to Cs and *T* via Equation (Equation 5). The units for *k* and $k^{{}^{\circ}}$ are s${}^{-1}$ (frequency units), and these are related to each other, $\Delta_{des}U$, $k^{{}^{\circ}}$, and the Boltzmann constant ($k_{B}$) through an Arrhenius type relationship (Equation (Equation 5)). This frequency is called the attempt frequency, and it is of the order of crystal lattice atomic frequencies (≅10${}^{13}$ s${}^{-1}$) [35].
(5)$$r_{des}=-\frac{dCs}{dt}=kCs^{n}=k^{{}^{\circ}}Cs^{n}exp\left(-\frac{\Delta_{des}U}{k_{B}T}\right)$$
where *n* is the surface desorption rate order. It is assumed that all adsorbed molecules occupy identical sites on the surface and that they do not interact with each other. In TPM, there are two possible regimes of data acquisition: flash desorption and adiabatic (slow) desorption. Slow desorption is commonly used for TPM by TDS. Here, the vapor pressure is proportional to *r*${}_{des}$, and the heating rate ($\beta_{h}$ = dT/dt) used must be linear. Then, Equation (Equation 5) is transformed into Equation (Equation 6):
(6)$$p\left(T\right)\propto -\frac{dCs}{dt}\frac{1}{\beta_{h}}=-\frac{dCs}{dT}=\frac{k^{{}^{\circ}}Cs^{n}}{\beta_{h}}exp\left(-\frac{\Delta_{des}U}{k_{B}T}\right)$$

The desorption rate divided by the heating rate ($r_{des}$/$\beta_{h}$) increases at the beginning of the temperature ramp on sample heating, but decreases at the end of the temperature program because the adsorbate coverage is spent. The vapor pressure depends on *T* and has a maximum value at $T_{max}$ that is related to $\Delta_{des}U$, the desorption rate order (*n*), and $k^{{}^{\circ}}$. In the case of first-order kinetics [36], Equation (Equation 6) is derived and set equal to zero to find the maximum:
(7)$$ln\left(\frac{\beta_{h}}{T_{max}^{2}}\right)=-\frac{\Delta_{des}U}{k_{B}}\frac{1}{T_{max}}+ln\left(\frac{k^{{}^{\circ}}k_{B}}{\Delta_{des}U}\right)$$

To measure Cs with time and temperature, GAO measurements using polarized MIR light were used. IR reflectance spectra were measured at different $\beta_{sub}h$ and initial surface loadings ($Cs_{{}^{\circ}}$). The relationship between Cs, density (ρ), and thickness (*d*) is shown in Equation (Equation 8), and the relationship between ΔR/$R_{{}^{\circ}}$ (where *R* is the reflectance, ΔR is $R_{{}^{\circ}}-R$, and $R_{{}^{\circ}}$ is the baseline reflectance) and *d* is given in Equation (Equation 8) [37].
(8)$$d=\frac{Cs}{\rho }$$
(9)$${\left(\frac{\Delta R}{R_{{}^{\circ}}}\right)}_{s}≅1-R_{s}≅-8\pi d\nu Im\left(\epsilon \right)cos\left(\phi \right)\propto Cs$$
or
(10)$${\left(\frac{\Delta R}{R_{{}^{\circ}}}\right)}_{p}≅1-R_{p}≅-8\pi d\nu Im\left(\frac{1}{\epsilon }\right)sin\left(\phi \right)tan\left(\phi \right)\propto Cs$$
where $R_{s}$ is the reflectance with the component of the electric field vector $E_{s}$ oriented perpendicular to the plane of incidence, $R_{p}$ is the component of the electric field vector $E_{p}$ oriented parallel to the plane of incidence, ν is the frequency of vibration of some mode, ϕ is the angle of incidence, Im(ϵ) is the imaginary part of the dielectric constant of the substance deposited (or energy loss function), and Im(1/ϵ) is the imaginary part of the inverse dielectric constant of the substance deposited (or longitudinal optical energy loss function). Combing Equations (Equation 8)–(Equation 10), a relationship between ($\Delta R/R_{{}^{\circ}}$)${}_{s,p}$ and Cs is derived. Equations (Equation 9) and (Equation 10) can be well approximated for d<<λ, where λ is the IR wavelength, which is in the order of 10,000 nm or larger. Values of *d* between 1 to 100 nm are considered ideal values. Logistic sigmoid fits were obtained from plots of $\Delta R/R_{{}^{\circ}}$ vs. *T* for various modes and substances (see Equation (Equation 11)), where *A*, *B*, and $a_{{}^{\circ}}$ are constants. Next, the derivative of $\Delta R/R_{{}^{\circ}}$ with respect to *T* was
(11)$$\left(\frac{\Delta R}{R_{{}^{\circ}}}\right)={\left(\frac{\Delta R}{R_{{}^{\circ}}}\right)}_{s}+{\left(\frac{\Delta R}{R_{{}^{\circ}}}\right)}_{p}=\frac{A}{1+exp\left(a_{{}^{\circ}}\left(T-T_{max}\right)\right)}+B$$

## 3. Results and Discussion

### 3.1. Spectroscopic Signatures

Figure 2a–d shows the decay of the vibrational IR signals for the HEMs studied. IR spectra were recorded every 12 s using GAP spectroscopy. For TATP, the peak area ($A_{p}$) between 1330 and 1407 cm${}^{-1}$ was calculated for each spectrum. Two bands located in the wavenumber range of 1330 to 1407 cm${}^{-1}$ were selected because they are isolated and are relatively narrow in comparison with the others. These vibrations were assigned as the out-of-plane bending of the methyl group $\delta_{as}$(CH${}_{3}$) by Buttigieg et al. [38] Brauer et al. [39] assigned that combination to CCC asymmetric stretching and CCO bending. For 2,4-DNT and TNT, the prominent signal located at 1343 cm${}^{-1}$ was used. This signal was assigned to C–NO${}_{2}$ vibration coupled to C–N stretching [40,41,42] (Figure 2b,c). This band was used for monitoring the kinetic behavior of the nitroaromatic compounds. The range used to calculate the areas was 1324 to 1372 cm${}^{-1}$. $A_{p}$ for a characteristic MIR region has an exponential decay. Thus, a fit to a natural logarithm function in terms of *ln* (A−A${}_{\infty }$) vs. time was applied to determine the sublimation constants (*k*) for TATP, 2,4-DNT, and TNT from the slopes (see Appendix A). Figure 2d shows the decay of the IR signals for RDX. The behavior of the area for the band at 1264 cm${}^{-1}$ and the band at 1321 cm${}^{-1}$ (N–NO${}_{2}$ symmetrical stretching [43,44,45,46,47,48]) vs. time are approximately linear. However, the IR intensity decay vs. time is exponential for the 1593 cm${}^{-1}$ band (N–NO${}_{2}$ asymmetrical stretching [43,44,45]). To determine the true behavior of RDX sublimation, a calibration curve using multivariate chemometrics methods was obtained and used for the prediction of Cs. Next, Cs vs. time was plotted, and the exponential behavior was observed.

### 3.2. Determination of Surface Concentration and Thickness for RDX

Calibration curves were prepared with the GAP spectral data for RDX using a PLS regression algorithm [49,50,51,52,53] in the Quant2 software package by Bruker Optics OPUS (version 4.2). Samples with RDX surface loadings were prepared using a smearing method [26,54,55,56] at room temperature. Thirty-six standard Cs from 100 to 1000 ng/cm${}^{2}$ were used for the calibration curve. The Cs standards were verified using a high-performance liquid chromatography (HPLC) technique reported by Wrable-Rose et al. [57] (see Appendix A). The regions used for the analyses were 1000–1260 cm${}^{-1}$, 1314–1380 cm${}^{-1}$, and 1560–1634 cm${}^{-1}$. Vector normalization (VN) preprocessing was applied to the spectroscopic data. VN normalizes a spectrum by first calculating the average intensity value and subsequent subtraction of this value from the spectrum. Then, the sum of the squared intensities is calculated, and the spectrum is divided by the square root of this sum. This method is used to account for differences in samples thickness [58]. Cross validations were performed, and the root-mean-square errors of cross validations (RMSECVs), root-mean-square errors of estimations (RMSEEs), and correlation coefficient squared (R${}^{2}$) were used as criteria to evaluate the quality of the correlations obtained. The optimum calibration curve had an RMSEE of 6 ng/cm${}^{2}$, an RMSECV of 8 ng/cm${}^{2}$, an R${}^{2}$ of calibration of 0.9997, and an R${}^{2}$ of validation of 0.9993. The model was obtained from five loading vectors for the spectroscopic data, and the significance of the statistics was at the level of *p* = 0.0001. The limit of detection (LOD) was calculated according to Equation (Equation 12) [59,60,61], where Δ(α,β,g) is a statistical parameter that takes into account the $\alpha_{p}$ and $\beta_{p}$ probabilities of falsely stating for the *g* free degree, and the leverage, $h_{{}^{\circ}}$, quantifies the distance of the predicted sample at zero concentration level to the mean of the calibration set. Figure 3 shows the PLS model derived from the data and the ideal model (*y* = *x*).
(12)$$LOD=\Delta \left(\alpha_{p},\beta_{p},g)\right)RMSEE{\left(1+h_{{}^{\circ}}\right)}^{0.5}$$

The value obtained for the LOD with the PLS model was 22 ng/cm${}^{2}$; this value is very low but it is in the order of published in the literature for RDX from grazing angle [62]. A second model using classical least squares (CLS) regression (or linear regression) was used for comparison. The results are shown in Figure 3 where the peak areas at 1321 cm${}^{-1}$ were used for the regression. Peak areas are shown on a second y-axis in Figure 3. The correlation coefficient obtained was R${}^{2}$ = 0.9896, and the LOD value was 103 ng/cm${}^{2}$. This was calculated as three times the standard deviation of the intercept between slopes [63,64]. The LOD for this model is larger than the one obtained using PLS because the signal at 1321 cm${}^{-1}$ disappears almost entirely for a surface loading of 120 ng/cm${}^{2}$ (Figure 2d). This does not happen for the signal at 1594 cm${}^{-1}$, but a good linear CLS model could not be obtained for this signal. As shown in Figure 4, the predicted Cs from the PLS model vs. time exhibits exponential behavior. The value of k was determined for various temperatures (see Appendix A). When the surface concentration is low, several bands disappear. However, the vibrational signals in the range of 1400 to 1650 cm${}^{-1}$ are highly persistent. A red shift is observed when the concentration diminishes or approaches monolayer coverage. In particular, red shifts of 2 cm${}^{-1}$ for the band at 1321 cm${}^{-1}$ and 3 cm${}^{-1}$ for the band at 1268 cm${}^{-1}$ are observed (see Figure 2d). A larger red shift for the band at 1594 cm${}^{-1}$ is observed (6 cm${}^{-1}$), but it is not possible to determine it exactly because vibrational signals for water are present in this range.

At low surface concentrations, where RDX is near to monolayer coverage, the effect of the interaction of RDX with the surface is most noticeable. RDX can interact with the metal surface through the NO${}_{2}$ group. This weakens the NO bond, thus explaining the red shift as well as the smaller red shift exhibited by the N–NO${}_{2}$ symmetric stretching band. The NO${}_{2}$ group interacts with the surface, reducing its mobility. This adds another component to the reduced mass of the oscillator, causing a decrease in frequency. Only one peak at 1594 cm${}^{-1}$ is observed for this layer, but two bands are typically exhibited by bulk samples at room temperature (α phase polymorph of RDX). The peaks in the spectrum of bulk RDX are observed at 1574 and 1596 cm${}^{-1}$ (see Appendix A). It has been suggested that the β phase crystalline polymorph of RDX (β-RDX) [43] is present in this layer (see Figure 2d and Appendix A). α-RDX and β-RDX are conformational polymorphs [65] that can be differentiated using vibrational spectroscopy (Raman or MIR).

### 3.3. GAP vs. GAO Measurements for RDX

A validation of GAP for RDX was carried out by comparing the results from GAP measurements with those obtained using GAO. No significant spectral differences between GAP and GAO spectra were observed for the range of 1000 to 1600 cm${}^{-1}$ (see Appendix A). However, a difference in the instrument detection capability was observed. The calculated signal-to-noise ratios (SNRs) for measurements using GAP were much larger than the corresponding values obtained by calculations using GAO measurements, particularly when the number of scans is small. These differences in SNRs decrease when the number of scans is large. The signal at 1594 cm${}^{-1}$ was used to calculate the values of the SNRs, and the noise was calculated from root-mean-square (RMS) values for baseline levels in the 1900 to 2100 cm${}^{-1}$ region. GAP measurements show a higher SNR at a low number of scans because the area averaged is larger than that for GAO and spatial averaging of a large area tends to decrease the noise levels obtained (see Appendix A). This suggests that GAP may be used for measuring surface kinetic processes that require small time intervals (the time for one scan at a resolution of 4 cm${}^{-1}$ and 10 KHz scanning velocity is approximately 0.5 s).

### 3.4. Sublimation Enthalpies and Desorption Energies

Two different methods were used to determine $\Delta_{sub}H$, i.e., TGA and GAP, and a third was used to determine the desorption energy, i.e., TPM ($\Delta_{des}U$). Using the TGA approach, $\Delta_{sub}H$ was calculated from a linear fit of $-R_{g}ln$($\nu_{s}$) and 1/*T*, where the slope corresponds to the enthalpy of sublimation and $R_{g}$ is the ideal gas constant. This approach worked well for all cases except for TATP, because in this case the fit was not linear (see Figure 5). For TATP, a multiple regression analysis was performed, in which $-R_{g}ln$($\nu_{s}$) was related to a three-term fit: 1/*T*, ln(1/T), and a constant (see Equation (Equation 13)) [66]. Next, the derivatives with respect to 1/*T* for the simple linear models and for the multiple-terms model were obtained to calculate the $\Delta_{sub}H$ for the bulk phase of the HEMs studied (see Equations (Equation 14) and (Equation 15)).
(13)$$-R_{g}ln\left(\zeta \right)=a\frac{1}{T}+bln\left(\frac{1}{T}\right)+c$$
(14)$$\Delta H=-R_{g}\left(\frac{\partial ln\left(\nu_{s}\right)}{\partial \left(\frac{1}{T}\right)}\right)=-R_{g}\left(\frac{\partial ln\left(k\right)}{\partial \left(\frac{1}{T}\right)}+T\right)=\Delta_{sub}U+R_{g}T$$
where ζ is k or $\nu_{s}$. The model described by Equation (Equation 13) was evaluated using the *p*-value of the model; the parameters a, b, and c; and the correlation coefficient (R${}^{2}$). A value of *p* < 0.0001 was found for the parameter and models, indicating a high statistical significance for both TGA and GAP for TATP. The model in Equation (Equation 13) was used for TATP, but when this was applied for the other explosives, the p values indicated insufficient significance. This indicates a simple linear behavior of $ln(\nu_{s}$) vs. 1/*T*, and that the changes in heat capacity ($\Delta C_{p}$) are near zero (or that their values are within the errors of the experiment). This result indicates that the change in the heat capacity is more significant in the temperature range studied for TATP than for the other explosives.
(15)$$\Delta H\left(T\right)=a+bT=\Delta H_{mean}+\Delta C_{p}\left(T_{mean}-T\right)$$

The GAP and TGA data for TATP were evaluated using a nonrandom residual analysis for simple linear models, and a random residual for the model described with Equation (Equation 13). For the other explosives, a random residual trend for the simple linear models was observed. The advantage of the model, described by Equation (Equation 13), is that the first derivate with respect to 1/*T* is a linear function of *T*. Δ*H* can be calculated for every temperature in the range evaluated from Equation (Equation 15), where $\Delta C_{p}$ is the difference between the heat capacity of TATP in the gas phase and that in the solid phase ($\Delta C_{p}$ = $C_{p}$(solid) −$C_{p}$(gas) = −b), The uncertainties in Δ*H* in the media temperature should be lower than for every other temperature [67]. Uncertainties (σ) in Δ*H* were calculated from direct contribution from the model and the indirect contribution [68], calculated from the propagation of uncertainties [69] (see Appendix A).

TPM was used to obtain the energy of interaction between the HEM and the surface. There are three possible hypotheses for a substance deposited on a surface. First, if the energy of interaction ($\Delta_{int}U$) has a value comparable to $\Delta_{sub}H$ or lower, then the value of $\Delta_{des}U$ can be approximated by the sum of $\Delta_{int}U$ and ($\Delta_{sub}H$$-R_{g}$$T_{mean}$). Second, if $\Delta_{int}U$ is zero or very small, then $\Delta_{des}U$ is approximately $\Delta_{sub}H-R$${}_{g}$$T_{mean}$. Third, if $\Delta_{int}U$ is larger than $\Delta_{sub}H$, two decays of Cs should be observed by TPM; first, a decay of bulk coverage by sublimation followed by a second decay of the monolayer coverage. The values for the calculated thermodynamic parameters $\Delta_{sub}H$ and $\Delta_{des}U$$\Delta_{sub}H$ were calculate from $\Delta_{des}U$ + $R_{g}T$ are shown in Table 1. Three models were obtained for TATP from TGA measurements. The first model for the sublimation obtained from the rate of sublimation measurements at different temperatures used a Δ*T* of 5 ${}^{{}^{\circ}}$ C (see Figure 5; labeled as TATP_TGA_1; table included as part of the Appendix A). The rates were measured at 1 ${}^{{}^{\circ}}$C and 0.5 ${}^{{}^{\circ}}$C (see Figure 5). Labels used were TATP_TGA_2 and TATP_TGA_3, respectively. The table containing these results can also be found in the Appendix A. The TGA experiments for TATP were performed in triplicate to prove that it was not a simple linear case. The samples used came from two different syntheses, and the time difference between the two sets of experiments was six months. $\Delta_{sub}H$ at $T_{mean}$ = 37.82 ${}^{{}^{\circ}}$C was 83 ± 5 kJ/mol for the first experiment and 87 ± 3 kJ/mol and 86 ± 2 kJ/mol at $T_{mean}$= 43.00 ${}^{{}^{\circ}}$C and $T_{mean}$ = 37.80 ${}^{{}^{\circ}}$C, respectively, for the second and third experiments. The value of $\Delta_{sub}H$ for TATP using GAP was 140 ± 14 kJ/mol at $T_{mean}$ = 20.9 ${}^{{}^{\circ}}$C. These values are different to that obtained by TGA, but the $\Delta_{sub}H$ values obtained by TGA in the temperature range of 24 to 27 ${}^{{}^{\circ}}$C (calculated by Equation (Equation 15)) are statistically identical to $\Delta_{sub}H$ obtained by GAP (see Table 2 and Appendix A). This suggests that the interaction between TATP and the substrate is very weak and that sublimation is the main phenomenon involved.

The values obtained for $\Delta C_{p}$ by TGA and GAP are different. For TGA, the values are 1.21 ± 0.09 kJ/mol·K, 1.50 ± 0.04 kJ/mol·K, and 1.24 ± 0.04 kJ/mol·K, whereas the value for GAP is 8.6 ± 0.9 kJ/mol·K. It is possible that, in GAP, the SS surface can affect the measured values due to the heat transferred from the surface and the surroundings. This hypothesis is confirmed by the fact that at room temperature, the $\Delta_{sub}H$ values obtained by TGA and GAP are statistically identical, where heat transfer from the surface to the surroundings is almost zero. For TGA experiments, the temperature of the surroundings is equal to the sample temperature for all isothermal measurements, but for GAP, this only occurs at room temperature. This difference in $\Delta_{sub}H$ in the absence of $\Delta_{int}U$ is only observed for explosives where $\Delta C_{p}$ is statistically different from zero. This is confirmed by fitting the model of Equation (Equation 13) to 2,4-DNT, TNT, and RDX for the $\Delta C_{p}$ values obtained by GAP and TGA (see Table 2). There are no significant differences statistically between the values obtained by GAP and TGA because the values of $\Delta C_{p}$ for these explosives is in the order of the experimental uncertainties, although for RDX in GAP the crystalline phase is β (β-RDX) and in TGA is α (α-RDX).

The nonlinear behavior of TATP and the high value of $\Delta C_{p}$ can be explained by the difference in the values of $\Delta_{sub}H$ found in the literature (see Table 3). For the value of $\Delta C_{p}$ obtained from Equation (Equation 13), it was necessary for both methodologies to obtain many points of temperature and use a large range of temperatures. Data from the literature was used to obtain $\Delta C_{p}$ (see Table 3). For experiments at relatively low temperatures, $\Delta C_{p}$ is near to zero and the fit is linear. This can also be observed in Figure 5. The curvature is only prominent at high temperatures. The values of $\Delta C_{p}$ from Oxley et al. are comparable to the present results. This can be attributed to the fact that they used a small number of temperature values, which is required to obtain low uncertainties in $\Delta C_{p}$.

For 2,4-DNT and TNT, the materials deposited for GAP experiments do not exist in solid crystalline forms. Rather, they adopt metastable phases in the form of droplets (see Appendix A). The contact angle (CA) for the droplets is 42 ± 3${}^{{}^{\circ}}$ for 2,4-DNT and 35 ± 1${}^{{}^{\circ}}$ for TNT (see Figure 2biii,ciii), indicating that 2,4-DNT has slightly less affinity for the surface than TNT. The CA changes during the sublimation process. This can be explained from the microscopic viewpoint in that molecules that interact directly with the surface and neighbors cannot be desorbed to the gas phase as easily as molecules that are far away from the surface and that are able to pass directly to the gas phase, generating a change in CA without a change in the enthalpy of sublimation. This behavior is illustrated in Figure 2bii,cii). The mechanism is similar for 2,4-DNT and TNT. This is corroborated by the existence of an isokinetic temperature found for the plot ln(k) vs. 1/*T*. The value found for the plot was 666 K. The estimated $\Delta_{sub}H$ for the metastable forms are 91 ± 5 and 108 ± 6 kJ/mol for 2,4-DNT and TNT, respectively, assuming that the interactions between the HEM and the surface are negligible. A contribution from $\Delta_{int}U$ for 2,4-DNT and TNT cannot be ruled out, because the end of the sublimation of the droplets leaves a film of molecules that were interacting with the surface initially. However, the value of $\Delta_{int}U$ for this film should be too small to be measured by TPM with TDS using GAO. The TGA method for TNT and 2,4-DNT revealed $\Delta_{sub}H$ values for the crystalline phases of 95 ± 3 and 94 ± 3 kJ/mol, respectively. These values are close to the literature values (see Table 4). $\Delta_{sub}H$ for the crystalline phase is statistically similar to $\Delta_{sub}H$ for the metastable phase for TNT and DNT. The metastable form is described as a supercooled liquid [14]. It is possible to induce the transition from the metastable phase to crystalline phase by applying mechanical pressure. This process destabilizes the pseudo equilibrium of the metastable form and induces a change in state to the more stable crystalline form.

The size of the droplets depends on the Cs as generated in the smearing deposition method. A size distribution was obtained for different Cs values. A normal distribution was observed for TNT at all concentrations when alcohols are used as solvents in the deposition process. The distributions for 2,4-DNT are far from normal at low Cs. The distributions were obtained by capturing several images for a selected Cs and measuring the size of the droplets from the image obtained (see Appendix A). This analysis is important for the development of standards for solids deposited on substrates for use in explosives detection devices. This explains the higher RMSECV for 2,4-DNT than for TNT found by Primera et al. [54,55]. When RDX was deposited on the metal surface from isopropanol solutions, films were observed (see Figure 2diii and Appendix A). These films are made of β-RDX polymorph, different from the bulk solid. In bulk solid, α-RDX polymorph is observed. Several previous studies have reported that the β-RDX conformer is metastable [65] relative to the α-RDX conformer. The energy difference between the two conformations is less than 1 kcal/mol [44,75,76,77]. The β-RDX polymorph was also formed when the RDX sample was allowed to sublimate and condense on a glass slide and upon depositing the sample from solutions [45] of various solvents (acetone, methanol, or saturated isopropanol). This is supported by the MIR spectrum and discussed above. Sublimation of solid α-RDX and a β-RDX film were measured by TGA and GAP, respectively, and $\Delta_{sub}H$ for the α-RDX and β-RDX phases were obtained. The calculated value for α-RDX is 99 ± 3 kJ/mol (Table 1). The value obtained by the GAP method for RDX is 169 ± 5 kJ/mol. This value is $\Delta_{sub}U$ + $\Delta_{int}U$. TPM was used to obtain $\Delta_{int}U$. Measurement of R for five bands (1576 cm${}^{-1}$, 1534 cm${}^{-1}$, 1316 cm${}^{-1}$, 1268 cm${}^{-1}$, and 909 cm${}^{-1}$) in the spectrum of RDX at three different $\beta_{h}$ values were used to obtain T${}_{max}$ from the first derivate of a logistic fit (Equation (Equation 11) and red square inset in Figure 6). Next, $-R_{g}ln$($\beta_{h}$/$T_{max}^{2}$) vs. 1/$T_{max}$ was plotted to determine the value of $\Delta_{int}U$ from the slope of the fit (Equation (Equation 7)). $\Delta_{int}U$ was found to be 19 ± 1 kJ/mol, and $\Delta_{sub}U$ for β-RDX now becomes 150 ± 5 kJ/mol and $\Delta_{sub}H$ becomes 153 ± 5 kJ/mol. This value is larger than the corresponding value for α-RDX.

## 4. Conclusions

Sublimation enthalpies were measured using GAP for materials that do not interact strongly with the surface and where $\Delta C_{p}$ in the temperature range of the study is zero or negligible. This is not the case for TATP because of a high value of $\Delta C_{p}$ was observed. The unexpected, temperature-dependent value for $\Delta C_{p}$ for TATP may be related to its high sensitivity to heat, friction, and shock. Using GAP, it was possible to differentiate between different phases and conformations of the materials. Using MIR techniques, the residence time of materials on surfaces was monitored, and the rates of sublimation of the materials from the surfaces were measured. The superiority of GAP over GAO found in this work is based on the detection sensitivity due to the high coverage area used in GAP. Using GAP, it is possible to detect highly energetic materials on metallic surfaces at the macro scale in 0.5 s (1 scan). Different types of mechanisms for sublimation on surfaces were found for the HEMs studied. For TATP, the sublimation takes place from small crystals to groups of islets. For 2,4-DNT and TNT, sublimation occurs from droplets that are part of a metastable phase. RDX is sublimated from a seemingly uniform coverage layer formed on the surface. The type of crystal phase that the HEMs assume on the substrate depends on surface-adsorbate adhesion forces vs. adsorbate–adsorbate intermolecular forces. Interactions between the HEMs and the surface can influence the rate of sublimation from the surface. The presence of this substrate-adsorbate interaction is demonstrated by the shift in the vibrational signals of RDX upon interaction with the SS substrate. For the case in which the interaction forces between the explosive and the surface are weak, the desorption energy should be minor compared to the sublimation enthalpy measured by TPM. 

## Figures and Tables

**Figure 1 molecules-24-03494-f001:**
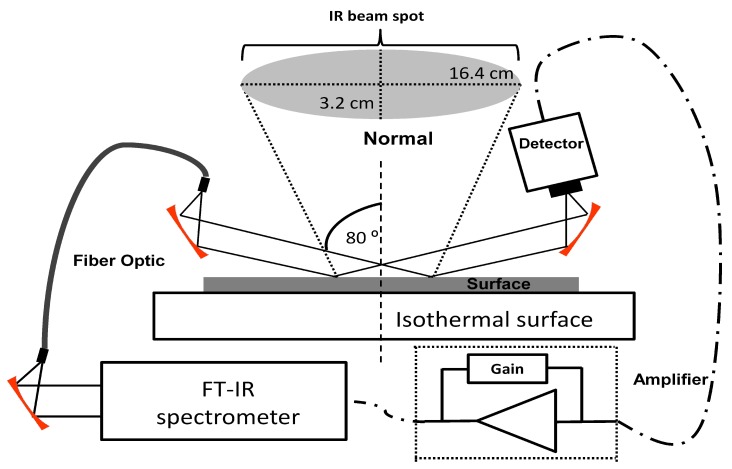
Grazing angle probe fiber-optic (GAP) experimental setup for the sublimation highly energetic material (HEM).

**Figure 2 molecules-24-03494-f002:**
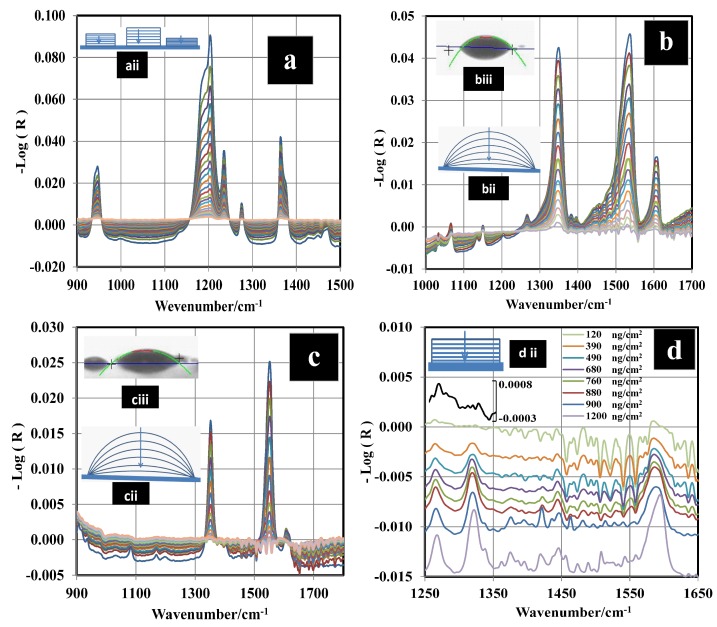
Mid-infrared (MIR) reflectance spectra for different HEMs undergoing sublimation on stainless steel (SS) surfaces. (**a**) TATP at 24 ${}^{{}^{\circ}}$C, (**b**) 2,4-DNT at 35 ${}^{{}^{\circ}}$C, (**c**) TNT at 70 ${}^{{}^{\circ}}$C, and (**d**) RDX at 80 ${}^{{}^{\circ}}$C.

**Figure 3 molecules-24-03494-f003:**
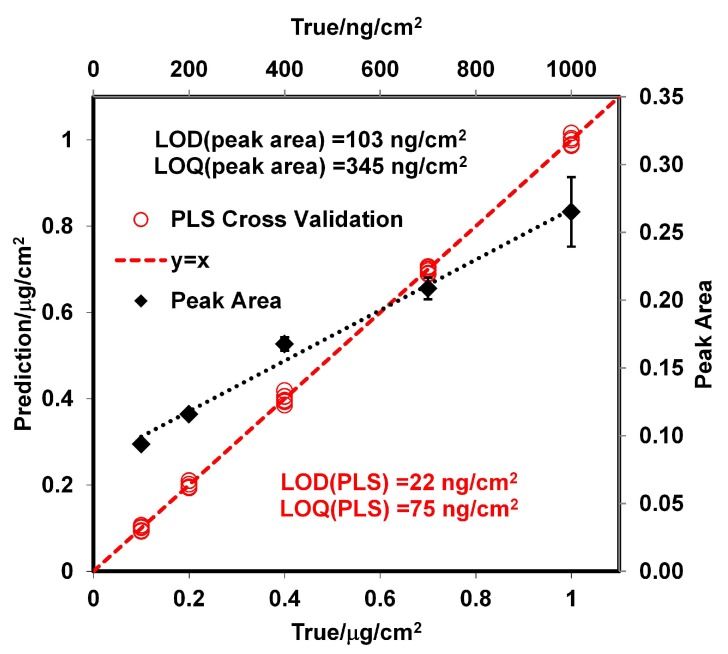
Calibration curves for RDX on SS by classical least squares (CLS) and PLS models.

**Figure 4 molecules-24-03494-f004:**
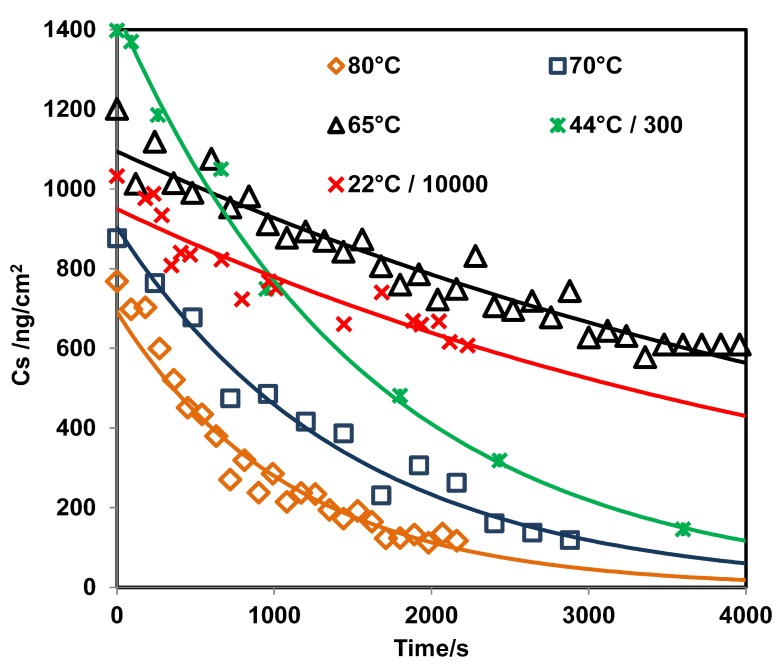
Prediction of Cs using the PLS model vs. time for RDX on SS at different temperatures.

**Figure 5 molecules-24-03494-f005:**
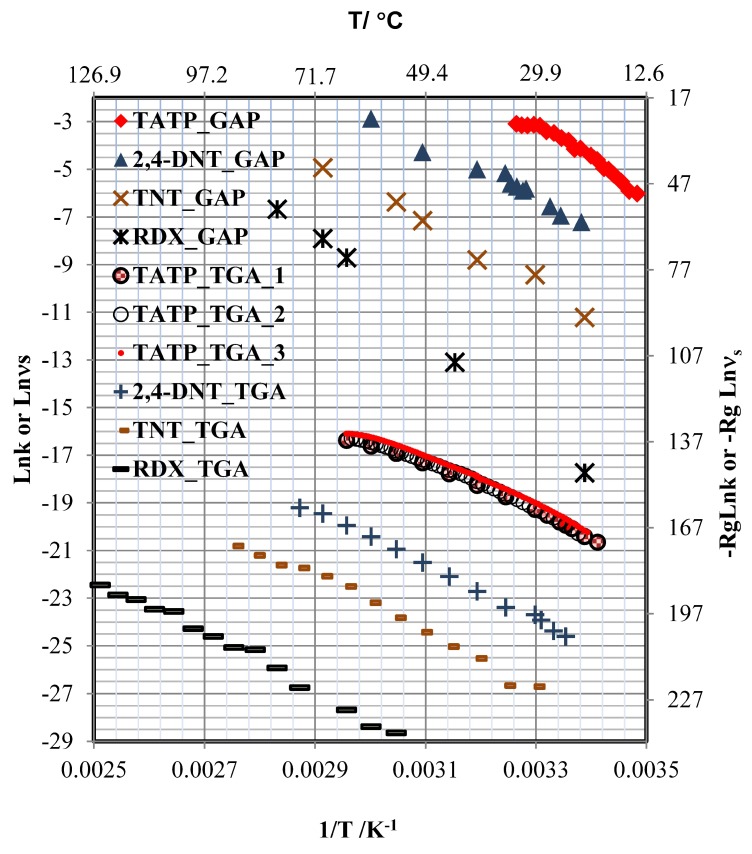
Arrhenius plots of GAP and TGA data used to obtain the sublimation rates for TATP, 2,4-DNT, TNT, and RDX. The units for GAP are s${}^{-1}$ and for TGA are kg·s${}^{-1}$.

**Figure 6 molecules-24-03494-f006:**
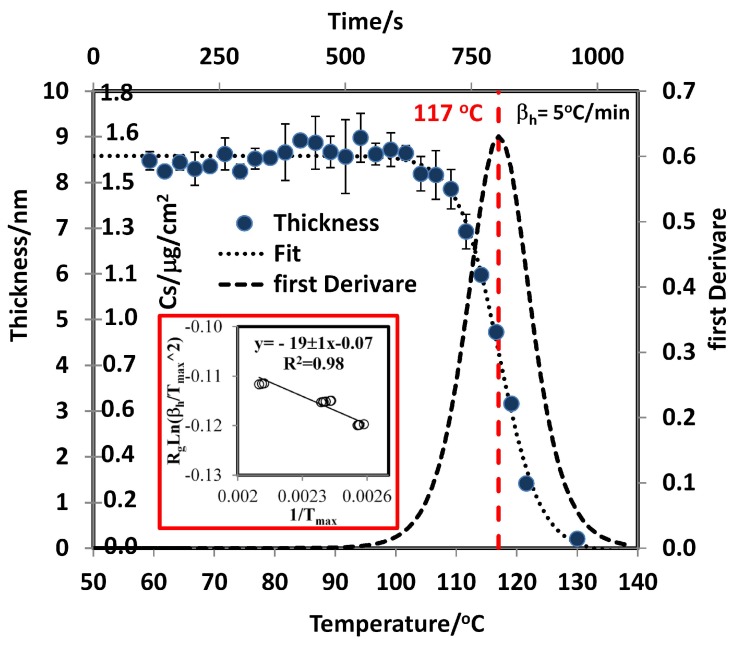
Plot of thickness vs. T(${}^{{}^{\circ}}$C) from TPM-TDS and d(ΔR/R)/dT vs. T.

**Table 1 molecules-24-03494-t001:** Enthalpy of sublimation for HEM and desorption energy for RDX on SS.

Exp	HEM	Range of *T*	$T_{\mathrm{mean}}$/${}^{{}^{\circ}}$C	$\Delta_{\mathrm{sub}}H$	R${}^{2}$
GAP	TATP	14–33	20.9	144 ± 14	0.997
TGA_1	TATP	20–65	37.5	83 ± 5	0.999
TGA_2	TATP	22–64	44.0	87 ± 3	1.000
TGA_3	TATP	21–63	37.8	86 ± 2	1.000
GAP	2,4-DNT	23–35	36.0	94 ± 5	0.986
TGA	2,4-DNT	25–75	46.6	94 ± 2	0.998
GAP	TNT	40–70	55.0	111 ± 6	0.998
TGA	TNT	40–65	52.5	95 ± 3	0.997
GAP	RDX	22–80	56.2	153 ± 5	0.998
TGA	RDX	55–125	90.0	99 ± 3	0.987
TPM	RDX	Δ*U*/kJ/mol	$\beta_{h}$/${}^{{}^{\circ}}$C/min	$T_{max}$/${}^{{}^{\circ}}$C	
		$\Delta_{int}U$ = 19 ± 1	5	117 ± 2	
			10	142 ± 3	
			20	200 ± 3	

**Table 2 molecules-24-03494-t002:** Enthalpies of sublimation for standards (kJ/mol) and $\Delta C_{p}$ (kJ/mol·K) for HEMs.

Molecule	$\Delta_{\mathrm{sub}}H^{{}^{\circ}}$ for GAP	$\Delta_{\mathrm{sub}}H^{{}^{\circ}}$ for TGA	$\Delta_{\mathrm{sub}}H^{{}^{\circ}}$ for TGA_2	$\Delta_{\mathrm{sub}}H^{{}^{\circ}}$ for TGA_3
TATP	104 ± 15	99 ± 6	107 ± 3	102 ± 3
2,4-DNT	103 ± 11	100 ± 5		
TNT	117 ± 31	120 ± 9		
RDX	165 ± 22 for β-RDX	112 ± 20 for α-RDX		
**Molecule**	**$\Delta C_{p}$ for GAP**	**$\Delta C_{p}$ for TGA**	**$\Delta C_{p}$ for TGA_2**	**$\Delta C_{p}$ for TGA_3**
TATP	8.6 ± 0.9	1.21 ± 0.09	1.50 ± 0.04	1.24 ± 0.04
2,4-DNT	1.1 ± 0.9	0.3 ± 0.2		
TNT	0.3 ± 1.0	0.9 ± 0.3		
RDX	0.4 ± 0.7	0.2 ± 0.3		

**Table 3 molecules-24-03494-t003:** Enthalpies of sublimation and $\Delta C_{p}$ values for TATP from this study and the literature.

	$T_{\mathrm{mean}}$ in ${}^{{}^{\circ}}$C	N of *T*	$\Delta C_{p}$ in kJ/mol·K	$\Delta_{\mathrm{sub}}H$ in kJ/mol
GAP	20.9	19	8.6 ± 0.9	142 ± 14
TGA_1	37.5	14	1.21 ± 0.09	83 ± 5
TGA_2	44	42	1.50 ± 0.04	87 ± 2
TGA_3	37.8	88	1.24 ± 0.03	85 ± 2
Damour et al., 2010 [69]	14.3	27	0	86.2 ± 1
Ramirez et al., 2006 [22]	50.0	7	0.75 ± 0.08	85.8
Felix et al., 2011 [70]	50.0	8	-	72.1
Oxley et al., 2005 [71]	40	6	0.3 ± 0.5	109
Oxley et al., 2009 [72]	32.2	7	0.6 ± 0.7	73
Dunayevskiy et al., 2007 [73]	0	-	-	81.3
Espinosa-Fuentes et al., 2015 [74]	46	32	1.5	103.8 ± 0.2

**Table 4 molecules-24-03494-t004:** Enthalpies of sublimation for DNT, TNT, and RDX in this study and the in literature.

HEM	Autor	$T_{\mathrm{mean}}$/${}^{{}^{\circ}}$C	$\Delta_{\mathrm{sub}}H$
DNT	This work from GAP	36	94 ± 5
DNT	This work from TGA	46.6	94 ± 2
DNT	Lenchitz 1970 [78]	64	98.3 ± 2.5
DNT	Lenchitz 1970 [78]	25	99.6 ± 2.5
DNT	Felix et al., 2011 [70]	52.4	96.2
DNT	Pella 1976 [79]	37	95.80 ± 1.25
DNT	Lenchitz 1971 [80]	–	99.6 ± 1.3
TNT	This work from GAP	55	111 ± 6
TNT	This work from TGA	52.5	95 ± 3
TNT	Edwards 1950 [81]	–	118.4
TNT	Dionne et al., 1986 [82]	25	113
TNT	Gershanik et al., 2010 [83]	40	97 ± 7
TNT	Oxley et al., 2005 [71]	36	137
TNT	Eiceman et al., 1997 [84]	114.5	87
TNT	Leggett 1977 [85]	26	141.1 ± 0.2
TNT	Hikal et al., 2014 [86]	67.5	95.9 ± 1
TNT	Hikal et al., 2011 [87]	–	100.2
TNT	Mu et al., 2003 [88]	32.5	131
TNT	Chickos et al., 2002 [89]	35	112.4
TNT	Cundall et al., 1978 [90]	25	113.2 ± 1.5
TNT	Felix et al., 2011 [70]	54.8	106.8
TNT	Pella 1977 [91]	–	99.2 ± 2.0
TNT	Lenchitz 1971 [80]	25	104.6 ± 1.7
TNT	Lenchitz 1970 [78]	65	103.3 ± 2.5
TNT	Jones 1960 [92]	–	118.4 ± 4.2
TNT	Hikal 2019 [93]	55	105.9 ± 1.4
TNT	Hikal 2019 [93]	55	102.1 ± 2.7
TNT	Hikal 2019 [93]	55	105.8 ± 1.6
TNT	Lee 2019 [94]	18.5	104.4 ± 2.4
RDX	This work from GAP	56.2	150 ± 5
RDX	This work from TGA	90	99 ± 3
RDX	Rosen et al., 1969 [95]	–	130.2
RDX	Gershanik et al., 2012 [96]	65	115 - 134
RDX	Eiceman et al., 1997 [84]	130	115
RDX	Hikal et al., 2011 [87]	–	128
RDX	Hikal et al., 2014 [86]	120	130 ± 2
RDX	Cundall et al., 1978 [90]	25	134.3
RDX	Felix et al., 2011 [70]	92	99.5
RDX	Chickos et al., 2002 [89]	–	112.5 ± 0.8
RDX	Rosen et al., 1969 [95]	77	130.1

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
