# Peer review of "Surface Persistence of Trace Level Deposits of Highly Energetic Materials"

_molecules, 2019, doi:10.3390/molecules24193494_

Round 1

Reviewer 1 Report

This is a fairly interesting article looking at various properties of explosives.  The novelty is the use of GAP to measure the properties.  There are no glaring issues and all my comments are relatively minor. 

A very general issue is that the article is hard reading at times.  Not because of the style (it is very well written) but the authors get tied up in the details and it can be distracting at times.  I feel they can shorten the article in parts, without affecting the outcome and it would be more impactful.  As a simple suggestion, I do not feel all the equations are necessary.  If they can just refer to a previous article they would not need all the details for defining the equations making it much simpler.  They have many equations already in the supplemental so if equations have been previously defined, all that text can go to supplemental.

Minor issues:

I am not sure this can easily be fixed but the use of acronyms is overwhelming.  I find myself having to go back and forth a lot which just makes the article hard to read.

Intro and abstract– double period.. ‘pressures by differential scanning calorimetry..’  I would expect that the abstract be worded a bit different rather than cut and paste.  The abstract is not concise.

Was there any check performed to verify the purity of TATP to exclude DATP for example. Could impurities be the reason for the non linear behavior?

Eq 1 (and 2), where did this come from? Is it instrument specific or sample specific? Is there a reference that the authors can provide? Maybe it is 27?  Just minor rewording so the equation does not appear out of thin air.

What was the nitrogen flow in the paragraph following equation 2?

Following equation 3, the * should be omitted in the units) and variables should be italic.  Variables throughout document need to be in ital.

Figure 2, can the authors use a finer line weight?  It would be easier to see and more dramatic.

Not totally necessary because of their agreement of data with the literature but if the authors did a comparison to a more well studied material, such as benzoic acid, those results should be included, even if in the supplemental.

Figure 4, I would suggest removing ‘day’ in the title since it was in thousands of seconds, not days.

Table 1 is not really useful.  Could move to supplemental or exclude that section.

Author Response

Dear Reviewer:

We have modified the manuscript according to your comments and detailed corrections are adding in the file

Reviewer 2 Report

The authors reported the sublimation behavior of trace HEMs by using the combination of tests and calculations, the results are fantastic and helpful for the area of energetic materials and public safety.  The data was well collected by GAP experiment and TGA tests. It will be better if HMX could be included in the samples since HMX shows similar structure and properties with RDX for better comparison. 

Author Response

We appreciate that you find the article interesting and It would be interesting to include HMX data this in the article, but we have decided to do this for future publication.